# A Non-Randomized Trial Investigating the Impact of Brown Rice Consumption on Gut Microbiota, Attention, and Short-Term Working Memory in Thai School-Aged Children

**DOI:** 10.3390/nu14235176

**Published:** 2022-12-05

**Authors:** Lucsame Gruneck, Lisa K. Marriott, Eleni Gentekaki, Kongkiat Kespechara, Thomas J. Sharpton, Justin Denny, Jackilen Shannon, Siam Popluechai

**Affiliations:** 1Gut Microbiome Research Group, Mae Fah Luang University, Muang, Chiang Rai 57100, Thailand; 2OHSU-PSU School of Public Health, Oregon Health & Science University, Portland, OR 97201, USA; 3School of Science, Mae Fah Luang University, Muang, Chiang Rai 57100, Thailand; 4Sooksatharana (Social Enterprise) Co., Ltd., Muang, Phuket 83000, Thailand; 5Department of Microbiology, Oregon State University, Corvallis, OR 97331, USA; 6Department of Statistics, Oregon State University, Corvallis, OR 97331, USA; 7Division of Oncologic Sciences, Oregon Health & Science University, Portland, OR 97331, USA

**Keywords:** gut microbiota, brown rice, cognitive performance, school-aged children

## Abstract

While dietary fiber has been shown to influence the composition of gut microbiota and cognitive function in adults, much less is known about the fiber-microbiome-cognition association in children. We profiled gut microbiota using quantitative PCR (qPCR) and evaluated cognitive function using the Corsi block-tapping test (CBT) and the psychomotor vigilance test (PVT) before, during, and after the dietary intervention of 127 school-aged children in northern Thailand. While we found that Sinlek rice (SLR) consumption did not significantly alter the abundance of gut microbiota or the cognitive performance of school-aged children, we did find age to be associated with variations in both the gut microbiota profiles and cognitive outcomes. *Gammaproteobacteria* was significantly lower in the control and SLR groups during the middle time points of both phases (Weeks 4 and 61), and its abundance was associated with age. Cognitive performance using CBT and PVT were also found to be age-sensitive, as older children outperformed younger children on both of these cognitive assessments. Finally, a multiple factor analysis (MFA) revealed that age and cognitive performance best explain individual variation in this study. Collectively, these findings further describe the influence of host variables on the microbial profiles and cognitive outcomes of school-aged children consuming Sinlek rice in Thailand.

## 1. Introduction

Human gut microbiota are highly diverse, and their composition can fluctuate throughout life depending on various factors such as genetic background, diet, and age [1,2]. Previous studies have reported a difference between the gut microbiota profiles of children and adults [3,4], with the degree of variation being higher among children than among adults [5]. As children develop, their gut microbial community matures to an adult-like configuration. During this developmental period, changes in the environment, especially diet, can have a significant impact on a child’s gut microbiota, suggesting that their gut microbiome is more malleable than that of an adult [6].

It is well recognized that dietary components including protein, fats, and carbohydrates have profound effects on gut microbiota composition [7]. A meta-analysis study has shown that dietary fiber intervention significantly altered the abundance of some beneficial bacteria (*Bifidobacterium* and *Lactobacillus*) and metabolites (butyrate) [8]. Rice is a rich source of carbohydrates [9] and a major staple food in Asian countries, including Thailand. Sinlek is a Thai rice variety (*Oryza sativa* L.) that is grown in the lowland regions and has a brown and grainy appearance. Brown rice offers numerous benefits over white rice, including being rich in phytochemical compounds and dietary fiber [10]. Consumption of brown rice has been shown to promote the growth of butyrate-producing bacteria in Japanese adults compared to white rice [11].

Accumulated evidence has highlighted important role of the gut-brain axis in influencing cognition and behavior [12]. Cognition refers to a process of acquiring knowledge and processing information, including attention, learning, memory, problem solving, and decision making [13]. Learning and memory are influenced by circulating hormones and neurochemistry [14,15,16], which can be impacted by food [17]. The gut-brain axis offers multiple mechanisms for food to influence brain homeostasis [12]. Gut microbes produce active metabolic compounds that facilitate communication between commensal gut bacteria and the brain [18,19]. For example, the microbial digestion of dietary fiber produces short-chain fatty acids (SCFAs) capable of indirectly stimulating the release of specific gut hormones from the enteroendocrine L cells, which can influence learning, memory, and mood. Fiber-rich brown rice has been shown to improve cognitive function in elderly populations receiving 6-month [20] and 24-month [21] interventions. Cognitive function impacts have been found as a result of microbiota transplantations in animal models [22,23,24] as well as with some intervention studies of probiotics/prebiotics [25]. However, a link between dietary intervention, gut microbiota, and cognitive function has only been elucidated in older adults [26,27]. To the best of our knowledge, no study has investigated such a relationship in school-aged children, including impacts on attention and short-term memory that are considered to be crucial cognitive abilities in school-aged children [28].

In this study, we investigated the effect of a fiber-rich Sinlek rice intervention on both the gut microbiota, using qPCR, and cognitive performance, using the Corsi block test (CBT; short-term working memory) and the psychomotor vigilance test (PVT; attention), of school-aged children using a non-randomized clinical trial conducted over 71 weeks. The associations between gut microbiota, cognitive performance, and demographic variables at each phase of the trial were explored in our study.

## 2. Materials and Methods

### 2.1. Study Participants

This study was a non-randomized clinical trial and included 127 children from the Ban Huai Rai Samakee elementary school in Chiang Rai, Thailand. Details on the recruitment of participants, ethical approval (Ethics Registry: REH-61204), metadata collection, and BMI categories have been previously described [29]. Figure 1 details a flowchart of the study design and data processing.

### 2.2. Study Design 

The study was divided into two phases and conducted at a rural elementary school during lunch periods. During the first phase (15 weeks), children in grades 1–3 (aged 6–8 years) were assigned to receive 100 g of traditional white rice, while children in grades 4–6 (aged 9–12 years) received Sinlek rice (SLR) (Figure 1). After pandemic-related school closures (Weeks 16–55), the second phase (Weeks 56–71) began, in which grades 1–3 received SLR and grades 4–6 received traditional white rice (WR). During the first four weeks of the intervention, SLR (50 g) was mixed with traditional WR (50 g) at a 1:1 ratio for adaptation. Traditional white rice was supplied by a local distributor in Chiang Rai, Thailand. The subjects had no history of acute or chronic inflammatory disease, no episodes of diarrhea, had not taken probiotic supplements in the month before sample collection, and had not been treated with antibiotics in the two months prior to sample collection. Sinlek rice (brown rice) was sourced from SOOKSATHARANA CO.,LTD (Phuket, Thailand). The nutritional components of the WR and SLR used in this study are shown in Table 1.

### 2.3. Cognitive Assessments

The cognitive performance of the school-aged children was measured using the CBT and PVT before, during, and after intervention. The computerized *Let’s Get Healthy!* platform [30] delivered cognitive tests to children in the Thai language [31], with the research team supporting the children’s use of touchpads during data collection sessions.

Short-term working memory was measured using a computerized version of the CBT, as described by Kessels et al. [32], which presented squares on a tablet screen that lit up one-by-one in a sequence (Amazon Fire HD 8 tablet, 8th generation, 2018). Corsi blocks were strategically fit to screen size so that participants could see all of the blocks at one time. Participants were presented with instructions and given repeatable practice rounds before the testing phase began (participant-initiated). Participants were asked to repeat the CBT sequence by tapping on the squares in the same order as was presented, with two opportunities to clear each level. The maximum block span for each participant was recorded, with larger numbers denoting longer working memory spans. The test took approximately 5–7 min to complete.

Attention was assessed using the PVT [33]. Use of the PVT has been well described for adults [34] and adolescents [35], as it provides an objective measure of reaction time (RT), errors of commission (i.e., false starts; where participants respond without a stimulus), and errors of omission (i.e., lapses; where participants do not respond in sufficient time). The elementary school children in our study were given the brief 3-minute version of the PVT (PVT-B), as described and validated by Basner et al. [36].

### 2.4. Quantitative Analysis of Fecal Microbiota

Fecal samples were collected from children at baseline, Weeks 4, 15, 56, and 61, and Week 71 in sterilized containers and were immediately frozen at −80 °C. Microbial DNA extraction and quantitative measurement of the absolute abundance of fecal microbiota were carried out as previously described [29]. Briefly, microbial genomic DNA was extracted from fecal samples using the innuPREP Stool DNA Kit (Analytik Jena Biometra, Jena, Germany), and DNA concentration and purity were determined using the Take 3 Micro-Volume Plate (Biotek, Winooski, VT, USA). The absolute abundance of microbiota 16s rRNA genes were then quantified through qPCR using Real-Time Thermal Cyclers CFX96 TouchTM (Bio-Rad, Singapore) based on the specific primers shown in Appendix A. The qPCR conditions and microbiota copy number estimations were preformed following the previous protocol [37]. The Log10 of 16S rRNA copy number per gram of wet weight feces is presented in our Appendix A.

### 2.5. Statistical Analysis

Data distribution was examined through the Shapiro–Wilk test and the Levene’s test (stats package version 4.0.3). The R package ggplot2 was used for data visualization [38]. Benjamini-Hochberg (BH) *p*-value correction was applied for multiple testing correction (*q*-value). Significance was determined at *q* < 0.05. All statistical analyses were performed in R software version 4.0.3 [39].

Differences in the mean absolute abundances of gut microbiota between the intervention groups for each time point were determined by *t*-test or Wilcoxon rank sum test according to the normality of the data distribution. Changes in gut microbiota between the time points of each phase (within subjects) were evaluated using pairwise *t*-tests or Wilcoxon signed rank tests following significant results from the one-way repeated measures ANOVA or the Friedman test (*p* < 0.05). Changes in the absolute abundances of gut microbiota between the weeks were expressed as log2 transformed fold changes (Log2FCs). Relationships between the absolute abundances of gut microbiota and age of school-aged children were determined using Pearson’s or Spearman’s rank correlation coefficient. The association between gut microbiota and intervention at each time point, adjusting for demographic variables (age, gender, BMI z-score, delivery mode, birth record, and ethnicity), was assessed through permutational multivariate analysis of variance (PERMANOVA) using the *adonis* function. To investigate the effect of intervention across the weeks of each phase, PERMANOVA was conducted using participants as strata and adjusting for covariates. The condition for homogeneity based on Euclidean distance was measured using the *betadisper* function. Permutation was performed at 999 in the R package vegan (2.6-2) [40]. We further determined the impact of intervention on gut microbiota using redundancy analysis (RDA) while adjusting for covariates. The treatments were used as constrained explanatory variables, and the absolute abundances of gut microbiota were used as response variables. The significance of constraints was assessed using an ANOVA-like permutation test (permutation = 999). A stepwise selection of explanatory variables was performed using the *ordistep* function based on AIC criteria.

Cognitive outcomes between the intervention groups for each week were compared using the Welch two-sample *t*-test or the Wilcoxon rank sum test. A difference in cognitive performance across the weeks of each phase (within subjects) was determined using the Wilcoxon signed rank test following significant results from the Friedman test (*p* < 0.05). We used RDA to determine the effect of intervention on the cognitive performance of school-aged children, while adjusting for age and gender. The effect of time points on the abundance of gut microbiota was also assessed through RDA using participants as strata. The RDA condition proceeded as described above. To determine the relationship between cognitive performance and gut microbiota, we used Spearman’s rank correlation coefficient.

Multiple factor analysis (MFA) was performed to explore the variations in gut microbiota that could be explained by the intervention and host variables (age and gender) using FactoMine R version 2.4 [41] and visualizing with Factoextra version 1.0.7 [42].

## 3. Results

### 3.1. Characteristics of School-Aged Children

After data quality control (Figure 1), we analyzed the remaining 85 and 57 children in Phases I and II, respectively. Children at the baseline were on average 7.02 and 10.52 years old for the WR and SLR groups, respectively. Baseline characteristics of the control and intervention groups were significantly different for age (*p* < 0.0001), weight (*p* < 0.0001), height (*p* < 0.0001), BMI z-score (*p* = 0.03), and feeding type (*p* = 0.02) (Appendix A). A similar pattern was also observed across the study time points, except for BMI, which was not significantly different between groups at Week 71 (Appendix A).

### 3.2. Effect of Sinlek Rice Intervention on Gut Microbiota of School-Aged Children

Considering each of the time points, there were no significant differences in the absolute abundance of gut microbiota between the control (WR) and Sinlek intervention (SLR) groups in the first phase of intervention (Phase I; baseline, Week 4, and Week 15). Multivariate comparisons using PERMANOVA (by margin) also showed that the intervention in Phase I had no significant effect on changes in gut microbiota after adjusting for demographic variables (Appendix A). We further determined an association between the abundance of gut microbiota and age and found that *Lactobacillus* was negatively correlated with age at the baseline (*Pearson’s correlation*; *R* = −0.24, *p* = 0.026) and Week 4 (*Pearson’s correlation*; *R* = −0.3, *p* = 0.002) (Figure 2b,d). For the second phase of intervention (Phase II), changes in microbial abundances, however, were marked at Weeks 61 and 71. The abundances of total bacteria (*q* = 0.032) and Firmicutes (*q* = 0.032) were significantly increased, whereas the level of *Gammaproteobacteria* (*q* < 0.0001) was significantly decreased following the SLR intervention at Week 61 (Figure 2e–g). These bacteria were also significantly associated with age, where the abundances of total bacteria (*Pearson’s correlation*; *R* = −0.32, *p* = 0.015) and Firmicutes (*Pearson’s correlation*; *R* = −0.32, *p* = 0.014, Figure 2h) decreased as age increased, while the absolute abundance of *Gammaproteobacteria* was positively correlated with age (*Pearson’s correlation*; *R* = 0.62, *p* < 0.0001, Figure 2i). At Week 71, the level of Bacteroidetes was significantly reduced in the SLR group (*q* < 0.0001) (Figure 2j) and its abundance was positively correlated with age (*Spearman’s rank correlation*; *R* = 0.57, *p* < 0.0001, Figure 2l).

Considering the changes in the abundance of gut microbiota across the weeks (within subjects) of the control and intervention groups, we observed similar patterns in the two groups of participants. *Gammaproteobacteria* seemed to highly fluctuate following intervention: its abundance was significantly decreased at Week 4 and Week 61 (Figure 3a,b, Appendix A). Levels of these bacteria appeared to bounce back, however, at the endpoint of each phase (Week 15 and Week 71). Moreover, the abundance of Bacteroidetes for the SLR group was lower at Week 71 compared to Weeks 56 and 61 (Figure 3a and Appendix A). A gradual decrease in the abundances of two microbial taxa, namely, *Ruminococcus* and *Bacteroides*, was also observed during Phase II, with their abundances being the lowest at the trial endpoint (Week 71) (Figure 3b and Appendix A), regardless of the intervention. Furthermore, we assessed whether there were differences in the abundances of gut microbiota between treatments and across the weeks using participants as strata. PERMANOVA indicated that the presence of treatment and/or the trial time point (week) during both phases had significant marginal effects on the abundance of gut microbiota (*p* < 0.05). The dispersions (variances) between groups and a model accounting for demographic variables with statistical significance are summarized in Appendix A. We further explored the strength of association and variation of gut microbiota explained by the intervention using RDA. Significant differences were found only in Phase II. The abundance of *Gammaproteobacteria* was more enriched in the control group (WR) than in the SLR group, while higher total bacteria, Firmicutes, and Bacteroidetes were associated with the SLR intervention at Week 61 (RDA1 explained 30.58% of total variance, *R*^2^_adj_ = 0.29, *p* = 0.001, Figure 3c). For Week 71, the abundance of Bacteroidetes was higher in the control group than in the SLR group (RDA1 explained 8.18% of total variance, *R*^2^_adj_ = 0.06, *p* = 0.005, Figure 3d). We then proceeded with a stepwise selection of explanatory variables (intervention and demographic variables) based on the AIC. The model revealed that only the intervention significantly explained the variations in the gut microbiota profiles of school-aged children (*q* = 0.035).

### 3.3. Cognitive Performance between the Control and Intervention Groups

In Phase I, we found significant differences between the control and intervention group in the cognitive outcomes. The SLR group showed significantly higher scores for on the Corsi block-tapping test (termed MMG, memory matching game) and overall performance (OVP) on the psychomotor vigilance test (PVT-B), while reaction times (RT) and lapses were lower than the control group (Figure 4). In Phase II, the patterns of cognitive performance remained unchanged. We also compared cognitive performance (within subjects) across the weeks of each phase and found that the control group (Phase I) performed better with regard to lapses at Week 15 compared to the baseline and Week 4 (*q* = 0.02) (Appendix A). RT was higher at Week 15 compared to Week 4 (*q* = 0.008), whereas no significant differences in MMG or OVP were detected. MMG was the only cognitive outcome that was significantly higher for the SLR group (Phase I) at Week 15 compared to baseline (*q* = 0.003) and Week 4 (*q* = 0.03) (Appendix A). In Phase II, there were no significant differences across the weeks in any of the cognitive performance measurements in either the control or SLR groups. Furthermore, age and gender were also included in the RDA, and the results showed no intervention effect on the cognitive performance of children in either group at any of the time points over the 71 weeks of the trial. Age, however, was found to significantly describe the variation of the cognitive outcomes at baseline (*p* = 0.008), Week 4 (*p* = 0.005), and Week 15 (*p* = 0.028) (Figure 5), for which age was positively correlated with MMG and OVP. No significant effects caused by the intervention, age, or gender on cognitive performance were found at Week 56, Week 61, or Week 71. When considering cognitive profiles across the weeks of each treatment in each phase, both week (*p* = 0.001) and gender (*p* = 0.001) significantly described variations in the cognitive performance of the control group (Phase I), for which the female sex and Week 15 were positively correlated with MMG and OVP (Appendix A). Children performed better at MMG at Week 15 in the SLR group (Phase I), while the female sex was positively correlated with lapses and RT (Appendix A). The effect of the treatment adjusting for age and gender was also assessed for each phase, while controlling for the week (time points). The RDA showed that age had a strong relationship with OVP and MMG in Phase I (Appendix A), whereas, in Phase II, performance with regard to RT and lapses was worse in the SLR group than in the control group (Appendix A). Overall, age and time points were more associated with cognitive performance of school-aged children than the intervention, and the intervention’s effect was only observed when constraining samples within each week of Phase II after adjusting for age and gender.

### 3.4. Association between Gut Microbiota and Cognitive Performance of School-Aged Children

We found weak correlations between gut microbiota and cognitive outcomes at all time points of the intervention based on the Spearman’s correlation coefficients. Several associations were significant (*p* < 0.05), but only one association, detected at Week 4 for the SLR group (*Roseburia* vs. Lapses, *rho* = −0.40, *q* = 0.04), remained significant after adjustment for multiple comparisons (Appendix A). A few associations that approached significance were found at Week 56 and Week 71. The associations identified in the SLR group at Week 56 included *Faecalibacterium*–RT (*rho* = 0.62, *q* = 0.07), *Prevotella*–OVP (*rho* = −0.55, *q* = 0.08), *Gammaproteobacteria*–OVP (*rho* = −0.58, *q* = 0.07), and *Faecalibacterium*–OVP (*rho* = −0.64, *q* = 0.05). At Week 71, *Faecalibacterium* was positively correlated with RT in the control group (*rho* = 0.62, *q* = 0.07).

Accounting for age and gender, a further analysis of the association between gut microbiota and cognitive performance at each time point was performed using an MFA to reveal variations between the individual profiles of the treatment groups (Appendix A). Although individual variation was explained by the abundance of gut microbiota in Dim 1 of both phases (*p* < 0.0001), a strong separation between the control and SLR groups was noted in Dim 2 (Figure 6 and Figure 7). The contrast profile between these two groups was influenced by age and OVP, which were positively correlated with Dim 2 (*p* < 0.0001), whereas both RT and lapses were negatively correlated (*p* < 0.0001). However, *Gammaproteobacteria* was the most influential in describing individual variation in this dimension at Week 61 (*p* < 0.0001). The level of this bacterium was lower in the SLR group (coordinate = −1.35, *p* < 0.0001). Despite unequal sample sizes, the MFA indicated that the difference in the profiles of school-aged children in this non-randomized clinical trial was more influenced by age and cognitive performance than by gut microbiota, suggesting less contribution by the intervention to individual variation in this study.

## 4. Discussion

Our study observed no significant changes in the absolute abundance of gut microbiota after the first phase of Sinlek rice intervention in school-aged children. Significant differences between groups, however, were noted in Phase II (Weeks 61 and 71) for total bacteria, Firmicutes, Bacteroidetes, and *Gammaproteobacteria*. We also observed significant differences in cognitive performance between the control and the intervention groups in both phases. We did not find a strong correlation between gut microbiota and cognitive performance at any time points. Further analyses suggested that, in this cohort, Sinlek rice intervention did not contribute to variations in the gut microbiota and cognitive profiles of school-aged children, as outcomes were primarily influenced by age.

Previous studies have shown that the absolute abundances of early-life gut microbiota significantly varied during the first two or three years of life [43,44]. In this study, the impact of age on gut microbiota profiles was also observed in school-aged children. Children receiving Sinlek rice exhibited increases in Firmicutes and decreases in Bacteroidetes abundances that negatively and positively correlated with age, respectively, in Phase II. It also appears that age influences the abundances of the above two major phyla, as their proportions in the gut have been found to shift throughout life. While Firmicutes and Bacteroidetes are the main phyla contributing to an adult-like gut microbiome structure during the first 4 years of life [45], their abundances were significantly varied when comparing children (9.8 years) to adults (≥40 years) [3], and adults (≥30 years) to the elderly (≥65 years) [46]. Our results suggest that these phyla might interact with age among school-aged children, irrespective of intervention. Although the abundance of *Gammaproteobacteria* was substantially lower for the SLR intervention group at Week 61 (2.2 times lower than the control group), we observed within each group a transient change in this class during Phases I and II. We hypothesize that a depletion of *Gammaproteobacteria* could be due to dietary shifts in children rather than a direct effect of the intervention, since the school reopened after months of closure due to the COVID-19 pandemic. Such a reduction in the abundance of this taxon, especially at Week 61, hinted that the class *Gammaproteobacteria* might be more sensitive to changes in diet (from home-cooked meals to school lunches) in younger children (aged between 7 and 8 years) as it correlated with age in our sample set. While the decline in the population of this class has not been established previously among school-aged children, a rapid response characteristic of the gut microbiota to altered diets might support our findings [47]. Moreover, the abundance of *Bacteroides* gradually decreased and was lowest at the trial endpoint among both children receiving white rice and Sinlek rice. A similar pattern was noted for *Ruminococcus*. We also observed a higher ratio of *Prevotella*/*Bacteroides* (*P*/*B*), which was negatively correlated with age (*rho* = −0.31, *q* = 0.02) at Week 71 (Appendix A). These bacteria are the predominant enterotypes in the human gut microbiome [48], and their composition can be altered by dietary components in the short term [49,50,51]; however, changing an enterotype’s status would require long-term dietary intervention [52]. Further, numerous studies have demonstrated that *Prevotella* and *Ruminococcus* are associated with dietary fiber, whereas *Bacteroides* dominates the gut of individuals consuming a Western diet [53,54]. The abundance of these taxa among school-age children might therefore be modulated by dietary changes, as previously mentioned.

Recent studies focusing on weight loss interventions have shown that changes in microbial profiles are associated with weight loss and an increase in the level of *Akkermansia*, regardless of intervention types [55,56,57]. In this study, we observed a decrease in BMI z-score in a few samples following the SLR intervention in Phase I. One sample (BH210) was obese at the baseline and then lost weight, to be later classified as overweight at Weeks 4 and 15. The BMI z-score of the other two samples (BH249 and BH273) reduced from overweight at the baseline to normal at Weeks 4 and 15. When looking at the gut microbiota profile of these individuals, a decreasing trend was observed in most bacterial taxa in the BH210 sample, except for Bacteroidetes (Appendix A). The abundance of gut microbiota was also observed to highly fluctuate in the BH249 and BH273 samples (Appendix A). Although statistical analysis was not possible due to the small number of samples, it would be interesting to study whether the SLR intervention effects changes in BMI, and its interaction with the gut microbiota in larger sample sizes.

While previous studies showed that animal source food intervention [58] and high protein consumption [59] are associated with a greater degree of cognitive ability in school-aged children, the use of Sinlek rice intervention was not found to have an effect on children’s cognitive performance, excepted in Phase II, during which the SLR group outperformed the control group (controlling for week), despite having a smaller sample size. Moreover, the cognitive outcomes which displayed an age-dependent pattern were consistent with prior reports of improved function with age on both the CBT [60] and PVT [61]. Repeated administration of PVT-B has not been shown to change PVT outcomes, including response time and lapses [62]. In our study, older children (aged between 9 and 12 years) performed better on all cognitive assessments compared to younger children (aged between 6 and 8 years). Our data demonstrated that older children in this study had greater memory span capacity (higher MMG) and better attention (lower RT and lapses) than younger children. Moreover, when age and cognitive outcomes were integrated with the gut microbiota, the first two variables were highly correlated. Such a relationship might be due to brain development in school-aged children, with no relation to changes in the gut microbiota. The CBT is a measure of visuospatial short-term working memory [63], which is often associated with hippocampal function [64,65]. The hippocampus is known to develop with age, with associated age-related improvements in memory (Riggins et al., 2018). Attention engages multiple brain regions, including the prefrontal cortex, motor cortex, and basal ganglia [66], regions that are also sensitive to age-related development. The basal ganglia are not fully developed in childhood and decrease in volume between the ages of 7 and 24 years [67], similar to other cortical regions that continue to develop throughout adolescence [68,69]. Our results confirm prior findings that cognitive outcomes improved as children developed, with this study adding contextual information about the microbial communities present in our adolescent population.

Consistent with previous findings [20,21], we hypothesized that brown rice may help improve cognitive abilities (attention and short-term working memory) in the understudied population of school-aged children. Our randomized sampling design, with clinical assessment and identification of biomarkers [70], allowed us to monitor more closely how brown rice intervention could provide a beneficial effect on cognitive function, including the potential impacts on mental health that were not included in the present study. Comparison between children and adults in the future would also be worthwhile.

Both cross-sectional and intervention (e.g., probiotics and prebiotics) studies have demonstrated that gut microbiota can influence cognitive health, with most findings indicating an improvement in cognitive outcomes as well as highlighting communication between the gut microbiota and the brain [71]. One study showed that individuals with high adherence to the Mediterranean diet have a high abundance of SCFA-producing bacteria, including *Faecalibacterium* and *Roseburia*. Both of these taxa were positively correlated with cognitive assessments such as BabCock memory and constructional praxis [72]. Here, we observed a negative relationship between *Roseburia* and lapses in the SLR group at Week 4, but this association was not maintained over time. *Faecalibacterium*, however, showed a positive relationship with RT despite treatments. Although these two genera are dominant butyrate-producing bacteria, their abundances might have a different effect on human cognition, as previously observed in patients with cognitive impairments [73,74]. Furthermore, an intervention dose of Sinlek rice, which was provided in a 1:1 ratio with white rice, may therefore not be sufficient to influence either the gut microbiome or cognitive performance of school-aged children. Future research using a full dose of Sinlek rice and studying metabolic profiles may help to unravel the complex relationship between gut microbiota and cognitive function.

*Lactobacillus* is one of the dominant bacteria found in breast milk, which can be transferred through breastfeeding [75]. In our study, we observed a negative association between *Lactobacillus* and age at the baseline and Week 4. More than 80% of children in the first phase of intervention were breast fed during infancy. An observed downward trend of this probiotic bacterium in older school-aged children implies that a decrease in the impact of breastfeeding could be a driving force during middle childhood.

Selecting an approach for microbiome profiling can be challenging, particularly when it is related to intervention, health, or diseases. The adoption of several approaches may add variability to the outcomes. Our quantification approach (qPCR), in particular, allowed us to determine the absolute abundances of gut microbiota and how much they changed following the intervention, whereas many studies utilizing 16S rRNA gene sequencing analyze microbial composition based on relative abundances [27,76,77]. Although the latter method aids in identifying the entire gut microbiome, interpreting compositional data generated by this method may make it difficult to identify the group of bacteria that are truly influenced by an intervention or health status [78]. Considering absolute abundance estimates of taxa may thus be beneficial in keeping track of the target bacteria and correlating their actual composition to studied conditions.

The main strength of this study is that it describes the interplay of a Sinlek rice intervention, the gut microbiota, and the cognitive performance of school-aged children, with age having a significant influence on both microbial profiles and cognitive outcomes. Nonetheless, several limitations need to be acknowledged. The small sample size and unequal numbers of subjects in the control and intervention groups may reduce the statistical power of our study. Comparisons of gut microbiota and cognitive performance within subjects overtime during the intervention could not be established due to missing subjects in Phase II with incomplete uptake of the intervention. Age, as a potential confounder, should be considered in future intervention studies. Although we focus on attention and short-term working memory, the cognitive assessments may need to be broadened (e.g., to include social function, planning, verbal tasks, and symbolism) to adequately describe the functional abilities of children, including mental health. Other variables that could have an influence on cognitive function, such as nutrition, wellbeing/socioeconomic status, and iron levels, were not collected due to language and cultural barriers, as the children were from various ethnic backgrounds. As our study involved children, intervention dosage was also a potential limitation. Moreover, it should be noted that the children’s diet outside of school hours was not controlled. The time between the two phases was also significantly extended due to the COVID-19 pandemic, and it was not possible to record the dietary patterns of children during that period. A recent study suggests that the pandemic caused some temporary changes in food consumption patterns [79]. As a result, there could be diet-induced variation on the microbiome or cognitive outcomes that may obscure any intervention effects that do exist. A metabolomics approach may help to clarify the connection between gut microbiota and cognitive function.

In conclusion, this non-randomized clinical trial revealed that Sinlek rice intervention did not significantly affect the abundance of gut microbiota or the cognitive performance of school-aged children. It did, however, find that age was significantly associated with variations in the abundance of gut microbiota and cognitive outcomes in both phases. Older children outperformed younger children on all cognitive assessments. A negative association between *Roseburia* and lapses was noted in the SLR group. Increasing the SLR dose or metabolic profiling would be needed to further understand whether Sinlek rice could exert a positive effect on the gut microbiota and improve cognitive function in children. Our findings indicate that age is directly related to gut microbiota profiles and cognition in school-aged children in northern Thailand.

## Figures and Tables

**Figure 1 nutrients-14-05176-f001:**
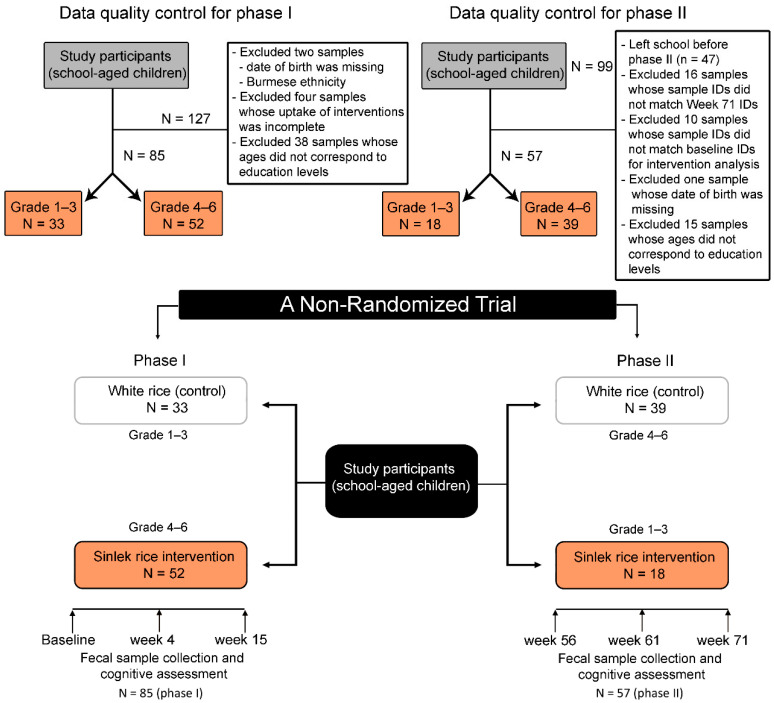
Progress of participants (school-aged children) through the non-randomized trial.

**Figure 2 nutrients-14-05176-f002:**
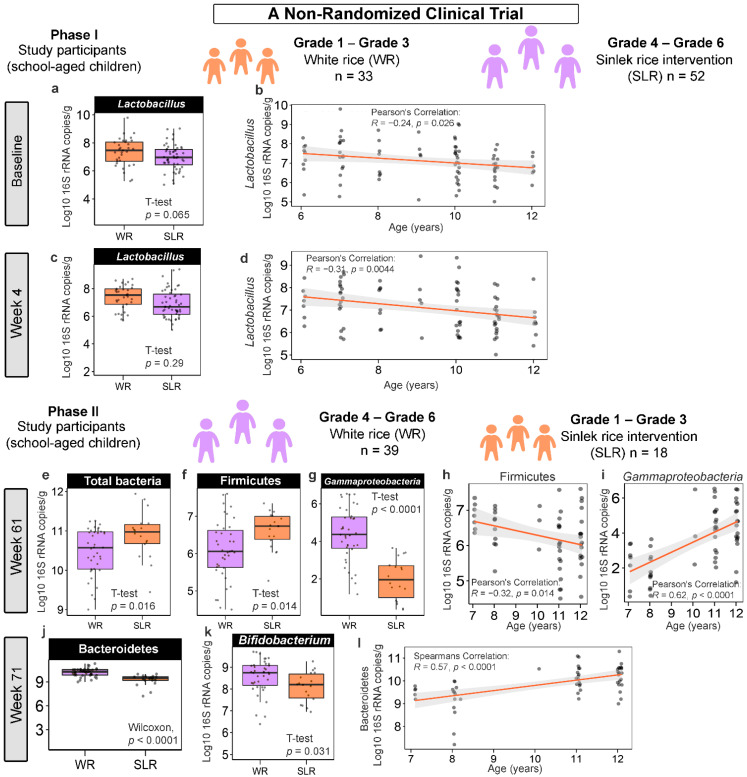
The ways in which Sinlek rice intervention affected the absolute abundance of gut microbiota of school-aged children in a non-randomized clinical trial. (**a**,**c**,**e**–**g**,**j**,**k**) Boxplots representing the normalized bacterial abundances at the phylum, class, and genus levels based on log10 qPCR 16S rRNA copy number per gram of feces. A difference between the mean absolute abundances of the control (WR) and intervention (SLR) groups was determined using either the two-sample *t*-test or Wilcoxon rank sum test. No significant differences between the mean absolute abundances of gut microbiota of the control and intervention groups at Week 15 (Phase I) and Week 56 (Phase II) were detected. (**b**,**d**,**h**,**i**,**l**) Associations between the absolute abundances of gut microbiota and age of school-aged children were determined using either the Pearson’s or Spearman’s rank correlation coefficient. WR, white rice (control); SLR, Sinlek rice intervention.

**Figure 3 nutrients-14-05176-f003:**
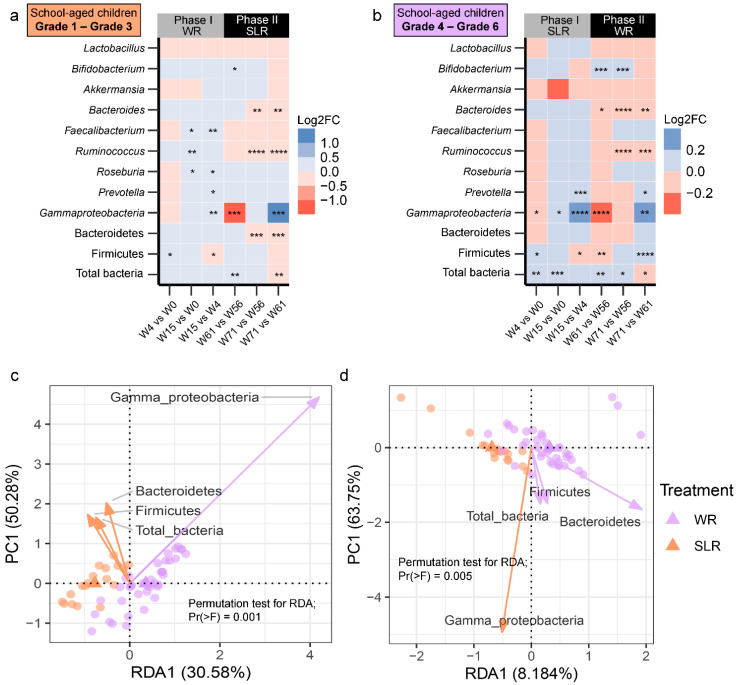
Differentially abundant gut microbiota across the weeks of intervention. (**a**,**b**) Heat plots showing the log2 fold change calculated based on the absolute abundances of gut microbiota between the weeks of the SLR intervention. Red indicates cases where the log2 fold change values were negative. Blue indicates cases where the log2 fold change values were positive. Differences in the mean absolute abundances of gut microbiota between the weeks were determined using either pairwise *t*-tests or Wilcoxon signed rank tests with Benjamini-Hochberg (BH) *p*-value correction following significant results from a one-way repeated measures ANOVA or the Friedman test (*p* < 0.05). **** *q* < 0.0001, *** *q* < 0.001, ** *q* < 0.01, * *q* < 0.05. (**c**,**d**) RDA plots for gut microbiota analysis using treatments at Week 61 (**c**) and Week 71 (**d**). Treatments (WR and SLR) were used as constrained explanatory variables and the absolute abundances of gut microbiota were used as response variables. Orange dots represent the control group (WR treatment), and blue dots represent the intervention group (SLR treatment). An orange triangle indicates the centroids of the WR group. A blue triangle indicates the centroids of the SLR group. Biplot arrows (gut microbiota) in the RDA plots of Week 61 and Week 71 were colored according to their associations with the treatment groups. The angle between a pair of vectors reflects their correlation. A significance of constraints was assessed using an ANOVA-like permutation test. WR, white rice (control); SLR, Sinlek rice intervention.

**Figure 4 nutrients-14-05176-f004:**
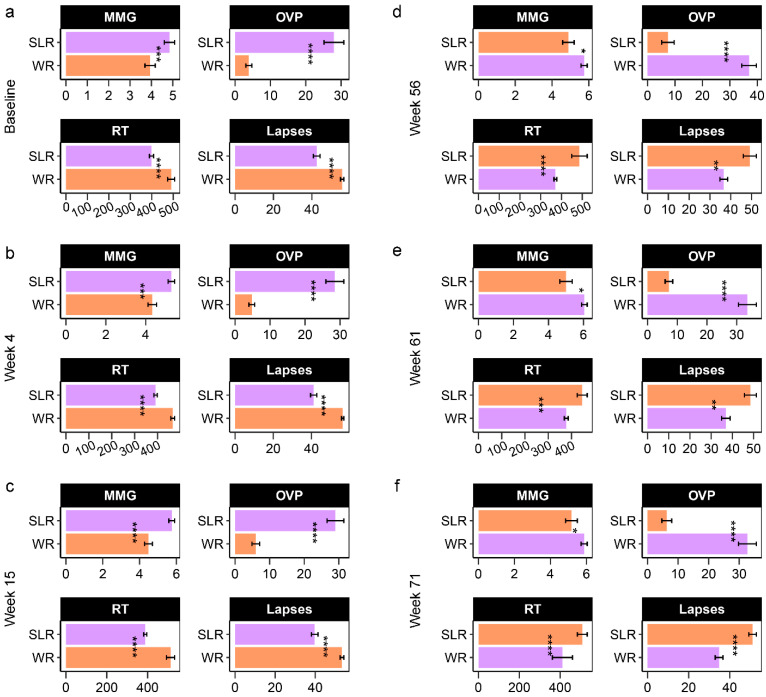
Barplots displaying the cognitive performance of school-aged children in a non-randomized clinical trial (Phase I: (**a**–**c**), Phase II: (**d**–**f**)). The difference in mean between the control (WR) and intervention (SLR) groups was determined using either the two-sample *t*-test or the Wilcoxon rank sum test with Benjamini-Hochberg (BH) *p*-value correction (*q*-value). **** *q* < 0.0001, *** *q* < 0.001, ** *q* < 0.01, * *q* < 0.05. MMG, memory matching game, which denotes Corsi block spans; OVP, overall performance (%); RT, reaction times (millisecond); lapses (millisecond). WR, white rice (control); SLR, Sinlek rice intervention.

**Figure 5 nutrients-14-05176-f005:**
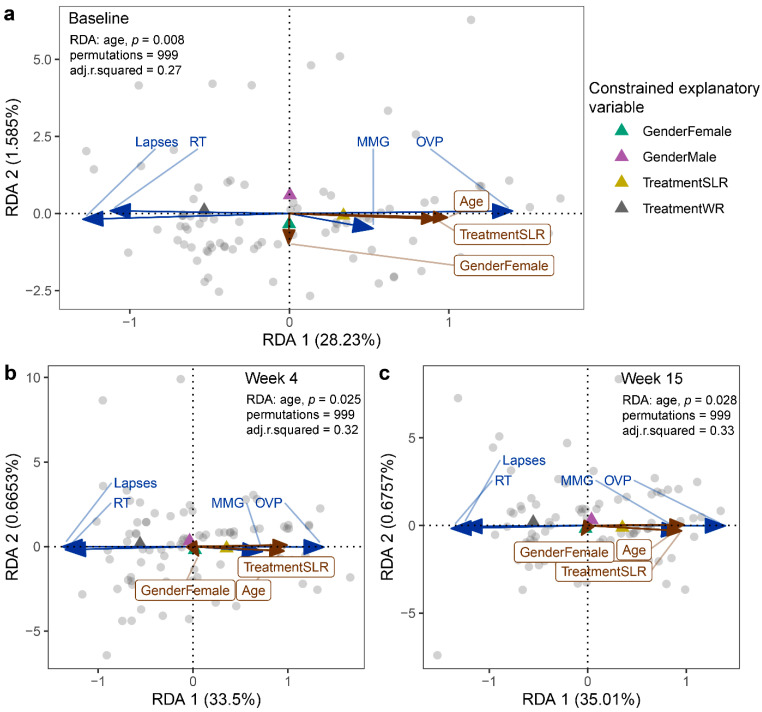
RDA plots displaying the effect of intervention on the cognitive performance of school-aged children at (**a**) the baseline, (**b**) Week 4, and (**c**) Week 15. Treatment, gender, and age were used as constrained explanatory variables, and cognitive performance was used as a response variable. Biplot arrows in the RDA plots represent cognitive performance (blue arrows) and constrained explanatory variables (brown arrows). A triangle denotes the centroid of each explanatory variable. The angle between a pair of vectors reflects their correlation. The significance of each constraint was assessed using an ANOVA-like permutation test. WR, white rice (control); SLR, Sinlek rice intervention; MMG, memory matching game; OVP, overall performance (%); RT, reaction times (millisecond); lapses (millisecond).

**Figure 6 nutrients-14-05176-f006:**
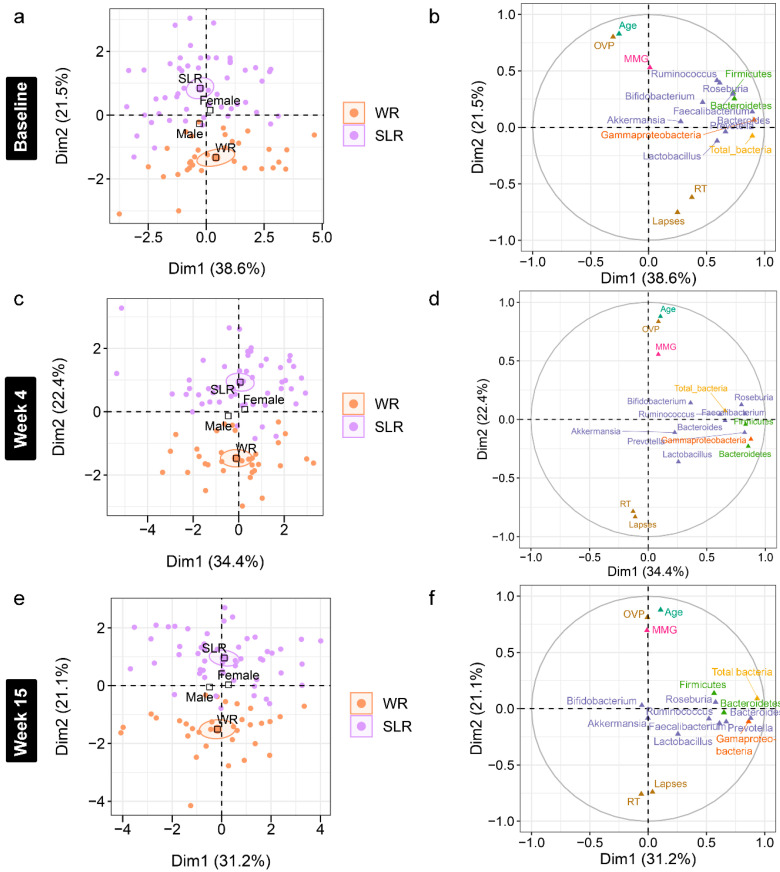
Multiple factor analysis (MFA) of variations in the gut microbiota and cognitive performance of school-aged children explained by the intervention, gender, and age in Phase I. (**a**,**c**,**e**) Individual factor maps broken down by the treatment groups (specified by the 95% confidence ellipses) in Dim 1 and 2. (**b**,**d**,**f**) Correlation between quantitative variables (gut microbiota at the phylum (green), class (orange), and genus (purple) levels, cognitive performance, and age) and dimensions (Dim 1 and 2). MMG, memory matching game, which denotes Corsi block spans; OVP, overall performance (%); RT, reaction times (millisecond); lapses (millisecond). WR, white rice (control); SLR, Sinlek rice intervention.

**Figure 7 nutrients-14-05176-f007:**
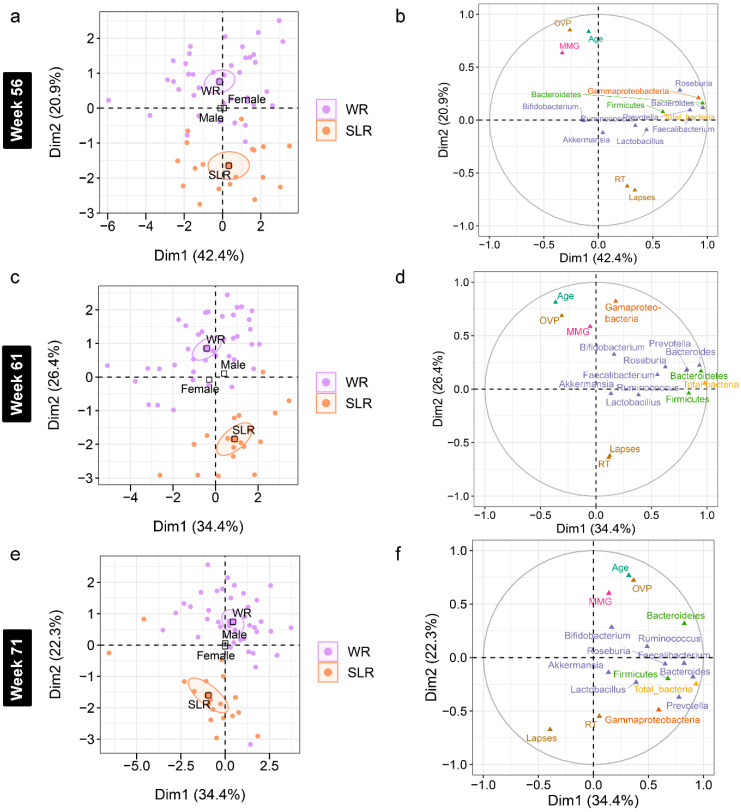
Multiple factor analysis (MFA) of variations in the gut microbiota and cognitive performance of school-aged children explained by the intervention, gender, and age in Phase II. (**a**,**c**,**e**) Individual factor maps broken down by the treatment groups (specified by the 95% confidence ellipses) in Dim 1 and 2. (**b**,**d**,**f**) Correlation between quantitative variables (gut microbiota at the phylum (green), class (orange), and genus (purple) levels, cognitive performance, and age) and dimensions (Dim 1 and 2). MMG, memory matching game, which denotes Corsi block spans; OVP, overall performance (%); RT, reaction times (millisecond); lapses (millisecond). WR, white rice (control); SLR, Sinlek rice intervention.

**Table 1 nutrients-14-05176-t001:** Proximate nutritional content of cooked brown rice (SLR) versus white rice (WR) per 100 g (g) serving.

Proximate Nutrients (g)	SLR	WR	*p*-Value	*q*-Value	Statistical Test
Ash	0.62 ± 0.01	0.64 ± 0.01	0.138	0.161	*t*-test
Moisture	60.68 ± 0.14	59.43 ± 0.15	0.000 *	0.000 *	*t*-test
Fat	0.41 ± 0.08	0.35 ± 0.09	0.385	0.385	*t*-test
Protein	3.20 ± 0.03	3.70 ± 0.05	0.080 *	0.112	Wilcox
Carbohydrate	34.38 ± 0.26	35.38 ± 0.11	0.004 *	0.014 *	*t*-test
Fiber (insoluble)	0.71 ± 0.19	0.51 ± 0.03	0.040 *	0.070	Wilcox
Resistance starch (soluble fiber)	1.10 ± 0.03	0.11 ± 0.02	0.040 *	0.070	Wilcox

The difference in the mean between WR and SLR was determined using the two-sample *t*-test or the Wilcoxon rank sum test (according to normality of the data distribution) with Benjamini-Hochberg (BH) *p*-value correction (*q*-value). A *q*-value of less than 0.05 is considered statistically significant (denoted by asterisk, *). We determined whether the median fiber and resistance starch (RS) contents of white rice were less than those of Sinlek rice using the two-samples Wilcoxon test. The statistical tests showed that Sinlek rice contained significantly more fiber and RS than white rice (*p* = 0.04). Values are presented as mean ± SD. WR, white rice; SLR, Sinlek rice.

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
