# Peer review of "A Non-Randomized Trial Investigating the Impact of Brown Rice Consumption on Gut Microbiota, Attention, and Short-Term Working Memory in Thai School-Aged Children"

_nutrients, 2022, doi:10.3390/nu14235176_

Round 1

Reviewer 1 Report

There are too many figures and too much text for the reader to understand the key (negative) messages/conclusions. This manuscript needs to be much more concise.

Cognitive assessment would be impacted by many other factors- nutritional well-being/socioeconomic status iron levels.

The title needs to reflect the study conclusions. 

Reviewer 2 Report

See the attached document

Round 2

Reviewer 1 Report

The authors are to be congratulated on the changes made which mean that the abstract, title and conclusions are easier to read and much more consistent with the data produced by the study